# Allograft Prosthesis Composite (APC) for Proximal Humeral Bone Loss: Outcomes and Perspectives

**DOI:** 10.3390/jpm13091301

**Published:** 2023-08-25

**Authors:** Pietro Gregori, Giovanni Perricone, Edoardo Franceschetti, Giancarlo Giurazza, Giuseppe Francesco Papalia, Pierangelo Zà, Rocco Papalia

**Affiliations:** 1Fondazione Policlinico Universitario Campus Bio-Medico, Via Alvaro del Portillo, 00128 Roma, Italy; pietro.gregori@unicampus.it (P.G.); giovanni.perricone@unicampus.it (G.P.); g.giurazza@unicampus.it (G.G.); g.papalia@policlinicocampus.it (G.F.P.); p.za@unicampus.it (P.Z.); r.papalia@policlinicocampus.it (R.P.); 2Research Unit of Orthopaedic and Trauma Surgery, Department of Medicine and Surgery, Università Campus Bio-Medico di Roma, Via Alvaro del Portillo, 00128 Roma, Italy

**Keywords:** shoulder replacement, shoulder revision surgery, allograft prosthesis composite, APC, humeral allograft, APC results

## Abstract

(1) Background: Allograft prosthetic composite (APC) represents one of the techniques used for reconstruction in large proximal humeral bone deficits. The present systematic review aimed at summarizing the state of the art of the technique and analyzing its outcomes. (2) Methods: The PRISMA guidelines were followed to perform this systematic review. A systematic electronic search was performed using PubMed (MEDLINE), EMBASE, and the Cochrane Library databases. All the studies analyzing the rates of allograft prosthesis composite were pooled, and the data were extracted and analyzed. (3) Results: A total of 10 studies were eligible for inclusion in this systematic review for a total of 239 patients. The rate of patient satisfaction with surgery was reported in 7 studies with a mean of 86.4% ± 13.64. The mean constant score was 45.7 ± 3.51, the mean ASES score was 63.58 ± 8.37, and the mean SST was 4.6 ± 1.04. The mean revision rate observed was 10.32% ± 3.63 and the mean implant survival was 83.66% ± 14.98. (4) Conclusions: Based on the currently available data, allograft prosthesis composite represents a valuable option for the reconstruction of proximal humeral deficits. All studies analyzed showed the favorable impact of this surgical technique on clinical outcomes and patient satisfaction.

## 1. Introduction

Proximal humeral bone loss (PHBL) may occur in cases of humeral resection for tumors, complex fractures, or following a revision of a failed shoulder arthroplasty [1,2,3]. Regardless of the etiology, the absence of the proximal humeral bone and the consequent lack of attachment of the capsule and the rotator cuff tendons limit the available effective alternatives for the management of this issue [4]. Traditionally, different reconstructive techniques such as arthrodesis [5], osteoarticular allografts [6], or tumor prostheses have been proposed. The main theoretical advantages of an osteoarticular allograft over a prosthesis are the supply of soft-tissue attachments with which reconstruct the remnant host soft tissues, lateralization of the pull of the deltoid, reconstitution of bone stock and improved contour of the shoulder. However, there are lots of complications associated with this type of reconstruction, such as subchondral collapse, fracture, infection, non-union, and late degenerative arthritis [7]. Although using an endoprosthesis avoids most of these complications [8], the prosthesis does not have the effective soft-tissue attachment sites of an allograft, and therefore it can be unstable, loose, or lead to fracture. Using an allograft prosthesis composite (APC) (Figure 1) combines the potential advantages of an allograft’s soft-tissue tendinous and capsular attachments with the benefits of a humeral prosthesis and theoretically avoids the disadvantages of each [7]. Modern reconstructive options for PHBL or resection include the use of a mega-tumor reverse prosthesis and reverse shoulder arthroplasty (RSA) with or without allograft augmentation [9]. Patients undergoing APC reconstructions of the proximal humerus seem to enjoy relatively good early function with respect to pain relief, active range of motion (ROM), and Musculoskeletal Tumor Society (MSTS) scores [10,11]. In particular, APC-RSA restores the lever arm of the greater tubercle and the deltoid “wrapping effect”, increasing ROM and implant stability while also restoring bone stock to spread out mechanical stress on the humerus, improving implant survival [12]. However, because there are no prospective or randomized trials, it is difficult to know which approach is best in terms of functional outcome, implant survivorship, or complications. Multiple previous studies have shown the difficulty associated with treating PHBL with APC [13,14]. Significant heterogeneity exists regarding the severity of bone loss encountered during surgery. There are factors that may predispose patients to more advanced bone loss, resulting in the use of larger allografts, longer stems, and additional modes of fixation [15,16]. Management of severe proximal humeral bone loss thus remains a surgical challenge. The purpose of our systematic review is to summarize the state of the art of the APC for humeral bone loss surgical technique and analyze its outcomes.

## 2. Materials and Methods

### 2.1. Information Sources and Search

An electronic systematic search of PubMed was carried out by two reviewers to identify eligible studies (Figure 2). Interrater reliability for study eligibility was measured using the statistic. The search was executed on 20 July 2023. Then, a manual search of the bibliography of each published study was performed, to find relevant articles that could potentially have been missed. Reviews, systematic reviews, and meta-analyses were also retrieved and read to broaden the search to include studies that might have been missed. The remaining articles were analyzed by two reviewers to exclude studies not fulfilling the eligibility criteria. The reviewers were not blinded to the authors, year, or journal of publication. Studies eligible for inclusion were categorized by study type, according to the Oxford Centre for Evidence-Based Medicine. The following categories were utilized: case reports, randomized controlled trials, case series, and cohort studies.

### 2.2. Data Collection Process

Two assessors independently extracted data from the eligible studies using a predefined data extraction procedure. For each study, we extracted data concerning the epidemiological characteristics of participants (age, sex) and the assessment of results (clinical outcomes, mean follow-up, complications, radiographic evaluation, revision rate). Data were analyzed using the R software (2020; R Core Team). The primary endpoint was the clinical outcome after APC in reverse shoulder replacement surgery. 

### 2.3. Quality of the Studies

The quality of the included studies was evaluated using the MINORS (Methodological Index for Nonrandomized Studies) score. The following domains were assessed: a clearly stated aim, inclusion of consecutive patients, prospective data collection, endpoints appropriate to the aim of the study, unbiased assessment of study endpoints, follow-up period appropriate to the aim of the study, loss to follow-up of <5%, prospective calculation of the study size, adequate control group, contemporary group, baseline group equivalence, and adequate statistical analysis (Table 1).

## 3. Results

A total of 23 studies were identified in the electronic search; of these, 10 were eligible for inclusion in this systematic review. No further studies were identified as relevant through the manual search. All the individual studies were retrospective. Study details are summarized in Table 1. The main indication was bone deficits after tumor resection or hemiarthroplasty/reverse shoulder arthroplasty failure. A total of 239 patients were included in the 10 studies. The mean age was 46.62 years (range: 6 to 87 years) (Table 2), with an average follow-up of 60.67 ± 19.55 months (Table 1). The average value was 11.7 on the MINORS scale.

### 3.1. Functional Outcome and Patient Satisfaction

The rate of patient satisfaction with surgery was reported in seven studies with a mean of 86.4 ± 13.64%. Different scores were applied to evaluate clinical outcomes. When reported, the mean constant score was 45.7 ± 3.51, the mean ASES score was 63.58 ± 8.37, and the mean SST was 4.6 ± 1.04. Regarding the range of motion of the shoulder, the mean forward elevation was 89.93° ± 11.86°, the mean external rotation was 21.1° ± 10.05°, the mean internal rotation was 4° or L4, and the mean abduction was 70.6° ± 10.09. 

### 3.2. Revision Rate and Complications

The mean revision rate observed was 10.32 ± 3.63%, and the mean implant survival, reported in two studies, was 83.66% ± 14.98 at 10 years follow-up. The most common complications were dislocation of the implant, non-union/resorption of the graft, periprosthetic fractures, and infections. (Table 3).

### 3.3. Radiological Assessment

Nine studies analyzed radiological assessments through the study of post-operative X-rays. The majority of these studies analyzed the incorporation of the bone graft with a mean of 86.4% (range 84 to 96%). The mean time to union of the allograft–host junction was 7 months (range, 3 to 13 months). Another radiological complication was the scapular notching, with a mean of 24.75% (range: 0 to 48%). The remaining radiological assessments are summarized. Because of heterogeneity and sensitivity to bias when not including randomized controlled trials, we narratively reported our results. Survival and complications are reported as proportions of the included patients. (Table 3).

## 4. Discussion

Revision shoulder arthroplasty has been more extensively performed in recent years for a variety of reasons, most commonly after a periprosthetic joint infection or mechanical failure. Moreover, the proximal humerus is the third most common site for osteosarcoma and the second most common site for all osseous sarcomas, most of them requiring surgery. While Ewing’s sarcomas and osteosarcomas occur characteristically in young adults and teenagers, chondrosarcomas occur in older individuals [22]. In patients undergoing wide excision of the shoulder girdle, this method has been used in tumor reconstruction to provide a functional upper extremity, although most of this literature focuses on the use of an anatomic humeral prosthesis [23]. Advocates of allograft prosthetic composites in RTSA address their benefits as the reattachment of the subscapularis tendon insertion, lateralization of the pull of the deltoid, reconstitution of bone stock, and improved contour of the shoulder [4,13]. In this setting, reconstruction of the shoulder after resection of a malignant or benign, locally aggressive primary bone tumor of the proximal humerus remains challenging. This is due to the increased bending and torsional forces on the humeral component when significant bone loss is present [18]. Moreover, in order to perform an adequate resection of the tumor, the deltoid musculature and joint capsule, and occasionally the rotator cuff, axillary nerve, glenoid, or scapula, may be included. Since there are no quality randomized prospective trials and there is no consensus on the best reconstructive technique after PHBL, reviewing the literature might be useful to evaluate the use of this treatment. Multiple risks of using allograft persist, including infection from the donor graft to the host, increased nidus for the de novo infection, graft resorption, cost of the graft, increased operative complexity and time, and non-union or malunion [19,20,21]. This systematic review aimed to evaluate the survivorship, as well as the clinical and radiological outcomes, of the allograft prosthesis composite (APC) technique (Figure 1) for the management of proximal humeral bone loss in shoulder arthroplasty. The main finding was that this technique has shown high survival (83.66% ± 14.98) up to ten-year follow-up, associated with an important increase in average ROM compared with pre-operative evaluation and also a high rate of patient satisfaction (86.4%). Even though the quality of the published evidence is low, the use of bone allograft support for the humeral revision construct demonstrates promising results. Furthermore, the high rate of satisfaction represents today a central variable for the decision-making process of the surgeon [24,25]. A direct comparison between APC reconstruction and no-bone-loss reconstruction techniques was performed in one study only [17]. They analyzed and compared patients with proximal humeral bone loss undergoing endoprosthetic replacement or a reverse shoulder replacement with or without the use of APC. Reverse reconstructions had improved American Shoulder and Elbow Surgeons scores (65 vs. 57; *p* = 0.01) and Musculoskeletal Tumor Society 93 scores (72 vs. 63; *p* < 0.001) versus hemiarthroplasty. Moreover, reverse reconstructions had improved forward elevation (85° vs. 44°, *p* < 0.001) and external rotation (30° vs. 21°; *p* < 0.001) versus a hemiarthroplasty. Teunis et al. [26] conducted a systematic review in 2014 comparing different proximal humerus surgical reconstruction techniques to identify which offered the best functional outcome measured by the Musculoskeletal Tumor Society (MSTS) score, had the longest survival rate, and had the lowest complication rate. They found that allograft prosthesis composites and prostheses had similar functional outcomes and survival rates, and both were able to avoid the fractures that were observed with osteoarticular allografts. These results must be viewed with caution since the small sample sizes and substantial heterogeneity of the patient populations hardly demonstrate the superiority of one technique over another. Excellent outcomes have been reported in other retrospective studies focusing on APC constructs of survivorship and the related satisfaction of the patients. Sanchez-Sotelo et al. [18] found an increased elevation of 41° to 98°, a mean ASES functional score at final follow-up of 66.1, a survival rate of 96%, and an overall survival rate of 80% at 5 years. Moreover, another retrospective consecutive case series of prospectively collected data from Cox et al. [20] found an overall survival rate of 88% at 5 years, as well as a humeral-sided survival rate of 94% at 5 years, 89% at 10 years, and 75% beyond 10 years, along with an increase in elevation from 49° to 75° and an increase in the ASES score from 33.7 to 51.1 at final follow-up. A meta-analysis of the different types of bone loss or techniques of APC could not be conducted due to the limited data in the literature. Regarding infection, the use of this technique is still debated. In their study, Boileau et al. performed nine two-stage APC procedures after failed arthroplasty. They found that the nine patients with preoperative infections, operated on in two stages, were considered to be cured at the last follow-up. They also found no significant differences in complications, reoperations, or clinical outcomes when they compared patients with postoperative infection vs. those without infection [9]. Sotelo et al. found that the infection and fracture rates in their study were relatively low, even if some patients had a history of infection or a malignant tumor requiring immunosuppressive therapy. They also did not find any statistical differences in both outcomes and complications rates between the primary and revision groups [18]. Chacon et al. performed their procedure on three patients who had a previous infection; they noted that two of them had some of the same outcomes and complications rates as the non-infected patients; the last patient with a previous infection suffered implant instability, but according to the author, this was caused by severe deltoid atrophy [4]. Finally, in their study, Cox et al. performed five APC procedures after failed HA for infection and two after failed RSA for infection, and they found the same outcomes and complications rate as the other patients [20]. Our study had several limitations. First, the high heterogeneity between the studies should lead to a critical interpretation of the results. The most relevant limitation is the low evidence level of the included studies, as most are level IV studies, and the low number of studies available on this topic, as well as the difficulty of comparing functional and satisfaction outcomes, which is related to the differences in study design. Moreover, the indications for APC were heterogeneous, and the absence of prospective studies with significant groups of controls decreased the quality of the data. Finally, a lack of subgroup analysis related to the different quantities of bone loss among the patients could produce a high risk of bias. This impedes the predictive value of our results for specific patients.

## 5. Conclusions

This study underlies high clinical improvements and low complication rates in patients treated with allograft prosthesis composite (APC) for proximal humeral bone loss. At the current state of research and with the highest number of studies analyzed on the topic, the APC technique represents a valid option for the treatment of these challenging cases. Further research and higher-quality studies should be carried out to individuate long-term outcomes and to compare this treatment with other techniques or implants in order to define where to find the right indication.

## Figures and Tables

**Figure 1 jpm-13-01301-f001:**
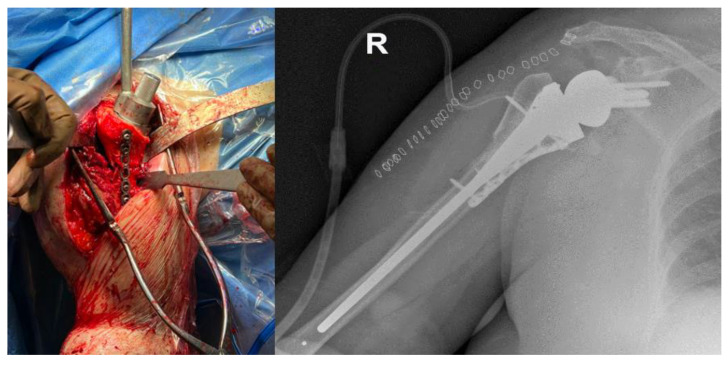
Intra-operative and post operative X-ray of Allograft Prosthesis Composite (APC) technique.

**Figure 2 jpm-13-01301-f002:**
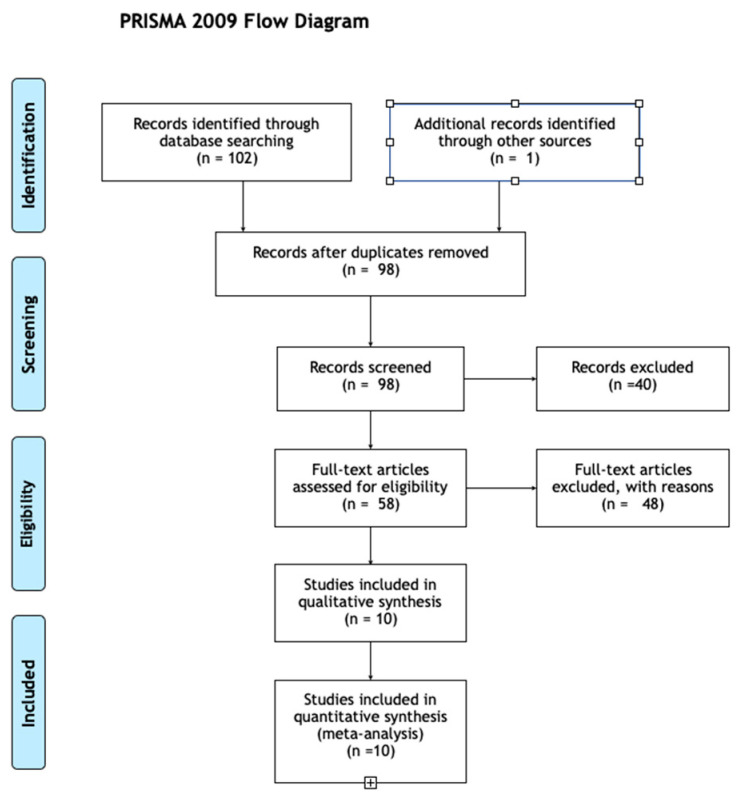
PRISMA flow chart of the study selection process.

**Table 1 jpm-13-01301-t001:** Study details.

Study	Year of Publication	Study Type	Level of Evidence	Mean Follow-Up (Months)	No. of Patients (Shoulders)	Women/Men	Mean Age	MINORS
Houdek et al. [17]	2020	Prospective study	III	84	10 (10)	5:5	57 ± 18	11/16
El Beaino et al. [9]	2017	Retrospective cohort study	IV	97	21 (21)	14:7	41 (20–80)	11/16
Chacon et al. [4]	2009	Retrospective cohort study	IV	30.2	25 (25)	23:2	-	12/16
Callamand et al. [11]	2020	Retrospective cohort study	IV	30	11 (11)	7:4	51 (19–87)	10/16
Abdeen et al. [7]	2009	Retrospective cohort study	IV	60	36 (36)	16:20	23 (6–74)	12/16
Sotelo et al. [18]	2017	Retrospective cohort study	IV	64	26 (26)	16:10	62 (33–86)	12/16
Black et al. [19]	2007	Retrospective cohort study	IV	55	6 (6)	4:2	40.7 (15–73)	12/16
Cox et al. [20]	2019	Case series	IV	67.9	73 (73)	55:18	67 ± 10	12/16
Boileau et al. [8]	2020	Case series	IV	48	25 (25)	16:9	59 (18–82)	13/16
Lazerges et al. [21]	2017	Case series	IV	70.8	6 (6)	2:4	65.5 (41–79)	12/16

**Table 2 jpm-13-01301-t002:** Main outcomes.

Study		Boileau et al. [8]	El Beaino et al. [9]	Chacon et al. [4]	Cox et al. [20]	Callamand et al. [12]	Black et al. [19]	Lazerges et al. [21]	Abdeen et al. [7]	Houdek et al. [17]	Sotelo et al. [18]
Constant Score	Pre	21 (5–47)	-	-	-	-	-	-	-	-	
	Post	42 (13–73)	-	-	-	49	-	46.1	-	-	
Patient overall satisfaction		76%	-	76% good/excellent 20% satisfactory 4% unsatisfactory	70% good/excellent 17% satisfactory 13% unsatisfactory	100%	-	81%	-	100% (RSA + APC) 75% (RSA)	7 excellent, 10 satisfactory, 9 unsatisfactory
Forward elevation	Pre	50		32.7	49	-	-	-	-	-	41
	Post	90	101 (1 y follow-up) 92 (5 y follow-up)	82.4	75	105	-	97	70 (deltoid intact) 59 (partial resection) 23 (total resection)	100 (RSA + APC) 76 (RSA)	98
External rotation	Pre	0	-	9.9	-	-	-	-	-	-	11
	Post	10	-	17.6	-	23	-	11	-	34 (RSA + APC) 27 (RSA)	31
Internal rotation	Pre	2	-	Sacrum	-	-	-	-	-	-	
	Post	4	-	L4	-	4	-	L4	-	-	
Abduction	Pre	-	-	40.4	45	-	-	-	-		
	Post	-	-	81.4	72	-	-	57	72 (deltoid intact); 52 (partial resection); 19 (total resection)	-	
VAS	Pre	6	-	-	-	-	-	5,1	-	-	
	Post	2	-	-	-	-	-	2,3	-	-	
ASES	Pre	-	-	31.7	33.8	-	-	-	-	-	
	Post	-	-	69.4	51.4	-	59 (last follow-up)	-	-	72 (RSA + APC) 61 (RSA)	66.1
SST	Pre	-	-	1.4	1.3	-	-	-	-	-	
	Post	-	-	4.5	3.5	-	-	-	-	6 (RSA + APC) 4 (RSA)	4.4

**Table 3 jpm-13-01301-t003:** Technique, survival, and complications.

Study	Cause of Proximal Humeral Bone Loss	Type of Technique	Revisions/Reoperations (%)	Implant Survival	Complications	Radiologic Assessment
Boileau et al. [8]	2 primary after tumor resection, 6 after failed MTP, 12 after failed RSA and 5 after failed HA.	15 step osteotomy 10 cement + screws 9 associated L’episcopo procedure	8 (32%) reoperation	-	- Instability: 6 - Loosening (glenoid): 4 - Infection: 2 - Allograft fracture: 1 - Temporary radial palsy: 1	Incorporation of the host: 24 (96%) scapular notching 12 (48%) NO humeral loosening
El Beaino et al. [9]	3 benign tumor 17 malignant tumor 1 renal cancer metastasis	7 cement + plate 14 cement	10.1% revision	-	- Loosening: 3 - Local recurrence: 2 - Periprosthetic fracture: 1 - Infection: 1	- Subluxation: 12 - Delayed union: 10 - Allograft resorption: 9
Chacon et al. [4]	24 after failed hemiarthroplasty 1 after failed bipolar hemiarthroplasty	Step cut osteotomy + cables 1 required plate for stability	-	-	- Dislocation: 2 - Allograft fracture: 1 - Non-displaced fracture of acromion: 1	- NO scapular notching - 1 humeral subluxation - Non-incorporation of the graft in the absence of resorption or fragmentation: 4
Cox et al. [20]	54 after HA (43 glenoid erosion and instability 3 tuberosity non-union 2 periprosthetic fracture 1 humeral stem loosening 5 infection) 17 after RSA 1 after anatomic 1 primary APC	Step cut + cables	-	88% (5 y) 78% (10 y) 67% (10 y+)	- Allograft fractures: 8 - Dislocations: 4 - Loosening: 5 - Glenosphere dissociation: 2 - Infections: 2 - 10 pz humeral loosening	- Radiographic incorporation: 53% (metaphysis) and 84% diaphysis
Callamand et al. [11]	6 chondrosarcoma 2 osteosarcoma 1 B-cell lymphoma 1 metastatic disease	4 fixation plate 7 self-stabilized Chevron osteotomy 5 associated with L’episcopo procedure	1 revision	-	- Dislocation: 1	- Humeral allograft consolidation: 73% - NO humeral loosening - Glenoid Allograft consolidation 100% - Scapular notching: 2
Black et al. [19]	4 chondrosarcoma 1 osteosarcoma 1 metastatic thyroid cancer	Chevron cut and cement	1 reoperation—1 revision		- Shoulder pain: 1 (did not follow indications)	- Radiographic non-union: 1
Lazerges et al. [21]	2 chondrosarcoma 1 Plasmacytoma 3 metastatic disease	Cemented	-	-	-	- Allograft host–junction consolidation: 5 - Non-union: 1 - Scapular notching: 2
Abdeen et al. [7]	19 high-grade osteogenic sarcoma 8 chondrosarcoma 3 metastasis 1 Ewing sarcoma 1 giant cell tumor 1 malignant fibrous histiocytoma 1 secondary high-grade osteogenic sarcoma 1 angiosarcoma 1 plasma cell myeloma	Cemented 5 + 31 Cemented + plate	3 revisions	88% at 10 y	- Dislocation: 1 - Loosening: 2 - Wound complications: 2	- Radiographic superior migration: 5 - Persisted radiolucent line: 1
Houdek et al. [17]	-	Cement		-	-	-
Sotelo et al. [18]	5 previous trauma, 3 tumor resection,	Cement, compression plate	80% reoperation rate, 2 revisions	96%	Hematoma with deep infection 1, dislocation 1, allograft fracture 1, periprosthetic fracture (distal to APC) 1, acromial fracture 1	Delayed union 1 Graft resorption 1

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
