# Peer review of "Allograft Prosthesis Composite (APC) for Proximal Humeral Bone Loss: Outcomes and Perspectives"

_jpm, 2023, doi:10.3390/jpm13091301_

Round 1
Reviewer 1 Report
Review note
This paper reviews an important topic – the outcomes of Allograft Prosthesis Composite (APC) for treating proximal humeral bone loss. The authors reviewed a nice set of literatures to address the topic, which will help further translational and clinical application. The structure is generally clear and logically organized. However, to grow into a publication, I think there are some issues the authors need to address.
1. Some abbreviations are not explained, for example CS in line 71.
2. I suggest the authors to intergrade the citation number in table 3 as well, just like what they do in table 1.
3. The hierarchy of evidence should be represented and discussed. Usually, randomized controlled trial >cohort study>case report.
4. This is not a mini-review, however, both introduction and discussion sections are too short for a regular-sized review, with only one paragraph for each. I recommend the authors to extend.
In general, this review is clear-cut and reflects an important issue. I hope the author(s) could find some of the above discussions helpful for improving the paper.
Minor editing of English language required
Author Response
We thank the reviewers for their comments and ideas to improve the quality of this work and to suggest the idealizations of the future ones. The specific responses to each comment were highlighted in yellow
- Some abbreviations are not explained, for example CS in line 71.
Correction has been made
- I suggest the authors to intergrade the citation number in table 3 as well, just like what they do in table 1.
Correction has been made
- The hierarchy of evidence should be represented and discussed. Usually, randomized controlled trial >cohort study>case report.
The Table 1 has been modified as requested, the different relevance of the studies has been discussed in the discussion.
- This is not a mini-review, however, both introduction and discussion sections are too short for a regular-sized review, with only one paragraph for each. I recommend the authors to extend.
Both the introduction and discussion have been extended as requested.
Reviewer 2 Report
• In the Introduction part, the contribution of the current study to the literature and the gap that the current study will fill in the literature should be mentioned.
• Figure 1 is mentioned in the text.
• The quality of Figure 2 should be increased.
• Figure 2 does not appear in the manuscript.
• Can the studies from 30 December 2022 to the present be included?
• In Conclusions, the importance of the current manuscript for future studies should be emphasized, and its superiority should be emphasized compared to the current “Reviews” in the literature.
• The study should contain original sentences and should be checked for comparison with the following studies: 10.1177/24715492231152143; 10.1002/jso.25061; 10.2106/JBJS.H.00815
Author Response
- Figure 1 is mentioned in the text.
Correction has been made
- The quality of Figure 2 should be increased.
Correction has been made
- Figure 2 does not appear in the manuscript.
Correction has been made
- Can the studies from 30 December 2022 to the present be included?
The research has been re-done. No further studies were eligible for the study. The date of the search has been updated.
- In Conclusions, the importance of the current manuscript for future studies should be emphasized, and its superiority should be emphasized compared to the current “Reviews” in the literature.
Conclusion has been modified as requested.